# Mechanism and Regulation of Microglia Polarization in Intracerebral Hemorrhage

**DOI:** 10.3390/molecules27207080

**Published:** 2022-10-20

**Authors:** Yuting Guo, Weibo Dai, Yan Zheng, Weilin Qiao, Weixuan Chen, Lihua Peng, Hua Zhou, Tingting Zhao, Huimin Liu, Feng Zheng, Peng Sun

**Affiliations:** 1School of Chinese Medicine, Shandong University of Traditional Chinese Medicine, Jinan 250355, China; 2Department of Pharmacy, Zhongshan Hospital of traditional Chinese Medicine, Zhongshan 528401, China; 3Research Center of Translational Medicine, Central Hospital Affiliated to Shandong First Medical University, Jinan 250013, China; 4Zhongshan Zhongzhi Pharmaceutical Group Co., Ltd., Zhongshan 528411, China; 5The Second School of Clinical Medicine, Guangzhou University of Chinese Medicine, Guangzhou 510006, China; 6School of Foreign Languages, Shandong University of Traditional Chinese Medicine, Jinan 250355, China; 7Department of Neurosurgery, The Second Affiliated Hospital of Fujian Medical University, Quanzhou 362002, China; 8Innovation Research Institute of Chinese Medicine, Shandong University of Traditional Chinese Medicine, Jinan 250355, China

**Keywords:** intracerebral hemorrhage, microglia polarization, inflammatory response, therapeutic target

## Abstract

Intracerebral hemorrhage (ICH) is the most lethal subtype of stroke, but effective treatments are lacking, and neuroinflammation plays a key role in the pathogenesis. In the innate immune response to cerebral hemorrhage, microglia first appear around the injured tissue and are involved in the inflammatory cascade response. Microglia respond to acute brain injury by being activated and polarized to either a typical M1-like (pro-inflammatory) or an alternative M2-like (anti-inflammatory) phenotype. These two polarization states produce pro-inflammatory or anti-inflammatory. With the discovery of the molecular mechanisms and key signaling molecules related to the polarization of microglia in the brain, some targets that regulate the polarization of microglia to reduce the inflammatory response are considered a treatment for secondary brain tissue after ICH damage effective strategies. Therefore, how to promote the polarization of microglia to the M2 phenotype after ICH has become the focus of attention in recent years. This article reviews the mechanism of action of microglia’s M1 and M2 phenotypes in secondary brain injury after ICH. Moreover, it discusses compounds and natural pharmaceutical ingredients that can polarize the M1 to the M2 phenotype.

## 1. Introduction

Intracerebral hemorrhage (ICH) is a common acute clinical cerebrovascular disease for which no effective treatment exists. ICH includes traumatic brain hemorrhage (TBI) and non-traumatic brain hemorrhage (ICH). Non-traumatic cerebral hemorrhage also includes intracerebral parenchymal hemorrhage, intraventricular hemorrhage, and subarachnoid hemorrhage, possibly due to primary or secondary causes. Among the risk factors for primary causes of ICH are hypertensive microangiopathy, cerebral amyloid angiopathy, coagulation disorders, drug use and other risk factors, while risk factors for secondary ICH include cavernous hemangioma, smoker’s disease, arteriovenous malformations, and aneurysms [1]. Brain injury ICH is mainly the primary injury caused by the expansion of the hematoma and secondary injury caused by the pathological reaction of the blood. The hematoma increases intracranial pressure causing mechanical compression of the surrounding brain tissue and decreasing blood flow to the brain, which may result in brain herniation in severe cases. In addition, the severity of the primary injury is related to the location and volume of the hematoma and the degree of edema in the brain tissue surrounding the brain injury. In contrast, secondary injury after brain hemorrhage is complex and includes inflammation, oxidative stress, excitotoxicity, and cytotoxicity [2]. Microglia, key innate immune cells within the brain, are thought to be the first cells to respond to various acute brain injuries, including brain hemorrhage. Previous studies have shown that microglia are activated by exudated blood components after ICH., and activated microglia are a major source of cytokines, chemokines, prostaglandins, proteases, and other immunomodulatory molecules in the brain that bind together to initiate the repair process secondary to brain injury [3,4]. Following an inflammatory response in the CNS, activated microglia and recruited macrophages present some common features (e.g., expression of shared phenotypic markers, ability to polarize to the M1/M2 phenotype, phagocytic behavior, and the amoeboid shape that activated microglia may acquire) [5]. Furthermore, it is difficult to distinguish; however, a broad M1 and M2 type classification remains a useful concept for understanding the functional status of microglia during the development of CNS injury and for exploring new therapeutic strategies [6]. Microglia around the lesion in the early stages of brain hemorrhage are activated and polarized into different phenotypes, including a pro-inflammatory phenotype (M1) and an anti-inflammatory phenotype (M2), which may interact with surrounding brain cells (e.g., neurons, astrocytes, oligodendrocytes) or may depend on the microenvironment in which they are located. Furthermore, the classical binary classification of microglia activated as pro-inflammatory and anti-inflammatory phenotypes is considered too simplistic. Microglia have overlapping functional states, which also need to be considered [7]. For example, M2b-like microglia can produce markers of both inflammation and tissue repair. It was found that, in a mouse model of ischemic stroke, both M1 and M2 phenotypes were present at the site of injury, and M1-related genes (inducible nitric oxide synthase (iNOS), CD1b, CD16, CD32, CD86) were upregulated from day 3 to day 14 after stroke; in contrast, mRNA expression of M2 markers (e.g., macrophage mannose receptor 1 (CD206), arginase 1 (Arg-1), IL-10, transforming growth factor-β(TGF-β)) on day one could be observed, peaking at days 3–5 and returning to pre-injury levels at day 14 [8]. However, in the ICH mouse model of bleeding within six hours, M1 microglia increased rapidly and decreased slowly over 14 days, whereas the M2 phenotype was at a low level on day one and increased by day 14 [3,9]. In addition, the M1 to M2 phenotype switch occurred within the first seven days after ICH, but the exact timing and predisposing factors of the phenotype change are unknown [10]. It was found that inhibiting the pro-inflammatory M1 response without completely eliminating it reduces secondary inflammation-related damage to nerve tissue and can accelerate the regression of damage. Because some inflammation is important to resolve nerve injury and promote repair, a completely dysregulated and persistent inflammatory state can instead limit tissue repair. This persistent and excessive neuroinflammation can lead not only to further tissue damage but also to poor neurocognitive outcomes in the recovery phase after nerve injury. Therefore, understanding the mechanisms involved in microglia during brain tissue damage and recovery is important for later prediction and identification of therapeutic targets [6,11].

## 2. Mechanisms in Activation of Microglia Following ICH

### 2.1. Activated M1 Microglia

After ICH, M1 microglia are activated to play a major role in phagocytosis and removal of necrotic neurons and cellular debris to reduce the release of harmful substances of inflammatory cytokines and chemokines [12]. However, as the number of M1 cells increases, phagocytosis decreases significantly and the secretion of inflammatory cytokines, chemokines, and other neurotoxic mediators increases, leading to extensive cellular damage [10]. In addition, protein hydrolases (e.g., metalloproteases) lead to disruption of the extracellular matrix and cellular integrity [13]. After brain injury, activated microglia are involved in oxidative stress and damage the blood–brain barrier; in addition, excess NO production from hemoglobin in the blood can also lead to the high permeability of the blood–brain barrier [14]. On this basis, the secretion of M1 microglia-specific chemokines leads to the infiltration and recruitment of blood-derived leukocytes, which exacerbates the inflammatory response and exacerbates neuronal death caused by excitotoxicity or oxygen-glucose deprivation [15]. Nuclear factor-κB (NF-κB) is a conventional transcription factor activated by lipopolysaccharide and regulates the expression of genes characteristic of most M1 phenotype microglia, namely pro-inflammatory cytokine genes [16]. In human brain tissue, NF-κBp65 was detected by immunohistochemistry to be expressed in the nucleus of glial cells [17], and was associated with elevated IL-1β and tumor necrosis factor (TNF). In clinical studies of perihematomal brain tissue, NF-κB was activated and migrated to the nucleus 13–48 h after ICH, and IL-1β and TNF were elevated within one day after ICH [18], These findings suggest a pro-inflammatory state early after ICH. In rodents, collagenase [19] and autologous blood [19,20] induced ICH models, and secretion of pro-inflammatory cytokines IL-1β, IL-6, and TNF were increased in the first three days [21], and returned to the normal range on day 7. The study shows that TNF receptor antagonist R-7050 attenuates neurovascular injury and improves function in mice with collagenase-induced ICH [22]. In addition, interferon gamma (IFN-γ) polarizes microglia to the M1 state and significantly increases the number of microglia expressing iNOS, thereby increasing the number of M1-polarized cells [23,24]. Studies have shown high expression of M1 markers such as CD16, CD32, and iNOS on microglia on days 1 and 3 after ICH [25], indicating that M1-like polarization of microglia occurs early in the acute phase of ICH. Most pro-inflammatory cytokine levels return to baseline 14 days after ICH [26]. However, it is unclear whether these cytokine levels change during the chronic phase of ICH and whether they affect the brain repair process.

In a collagenase and an autologous blood-induced mouse model of ICH, Toll-like receptor (TLR)2 and TLR4 proteins on microglia exhibited upregulation 6 h after ICH and remained high for the first three days [20,25], and increased expression of TLR2 and TLR4 proteins were detrimental to the prognosis of patients with ICH [27]. TLR4 can promote the inflammatory response by activating the NF-κB signaling pathway through the myeloid differentiation primary response protein MYD88, and the TIR structural domain-containing junction molecule 2 [28,29], and TLR4 antagonists can reduce the production of pro-inflammatory cytokines (IL-1β, IL-6, TNF) after intervention [30]. In addition, TLR2-TLR4 heterodimers trigger inflammatory responses, suggesting a role for TLR2 in brain hemorrhage similar to that of TLR4 [31]. It was found that the levels of IL-1β, IL-6, and TNF were significantly lower in TLR2 and TLR4 knockout mice after brain hemorrhage than in wild-type mice [20,31], thus supporting the link between TLR2 and TLR4 activation and M1 microglia. In addition, high mobility group protein 1 (HMGB1) was found to promote M1 microglia polarization by increasing TLR2 and TLR4 signaling [32]. HMGB1 is a classical damage-associated molecular pattern usually localized in the nucleus as a DNA-binding protein and involved in nucleosome formation and the stabilization of gene transcription. HMGB1 is upregulated in neural and immune cells after brain injury. It activates the myeloid differentiation primary response protein (MyD88)/NF-κB pathway downstream of TLR on the microglia surface, generating M1-type microglia polarization and releasing pro-inflammatory factors, leading to further neuronal damage [33]. Gao et al. treated TBI mice with the HMGB1 inhibitor glycyrrhizin (GL). They showed a decrease in the expression of M1 pro-inflammatory cytokines and their mRNA and an increase in the number of M2-type microglia [33].

### 2.2. Activated M2 Microglia

Microglia can promote cerebral edema and cellular debris clearance, promote cerebral angiogenesis, and improve neurological function through alternate activation of the M2 phenotype. Expression of M2 signature anti-inflammatory cytokines is also seen after ICH. IL-10 is secreted by microglia and macrophages and can induce differentiation of microglia and macrophages cultured in vitro to the M2c subtype [34,35], and enhance phagocytosis of monocytes [36]. In addition, IL-10 also induces production of suppressor of cytokine signalling (SOCS)1 and SOCS3, where SOCS3 inhibits macrophage pro-inflammatory cytokine production via signal transducer and activator of transcription (STAT) 3 [37]. STAT3, a member of the STAT family, is involved in various abnormal CNS pro-inflammatory responses. The STAT3 signaling pathway is involved in pro-inflammatory responses associated with the M1 phenotype, which may contribute to early brain injury in subarachnoid hemorrhage (SAH). It was shown that microglia-specific STAT3 deficiency upregulates the expression of major anti-inflammatory factors such as IL-4 and enhances anti-inflammatory function by triggering microglia polarization from M1 to M2, thereby reducing neuroinflammatory response after SAH and improving neurological dysfunction and neuronal apoptosis [38]. Studies have found elevated levels of IL-10 in the blood [39] and brain tissue of patients with ICH [40], and upregulation of early IL-10 expression correlates with rebleeding following ICH [41]. IL-13 and IL-4 are anti-inflammatory cytokines that promote M2 microglia polarization. IL-13 is secreted by Th2 cells and shares a receptor subunit with IL-4, which is secreted by Treg cells and Th2 cells, both signaling through the Janus activated kinase (JAK)-STAT6 pathway [42], and upregulates the expression of M2 microglia, including Arg-1 and chitinase-like protein 3 (YM1) in microglia type microglia marker expression [43,44,45]. In addition, IL-4 is thought to drive microglia polarization to the M2 type [23]. Studies have shown that intracerebral injection of IL-4 inhibits M1 microglia polarization and promotes M2 microglia, thereby improving recovery from neurological dysfunction after ICH [46]. IL-4 and TGF-β1 have anti-inflammatory effects [26,46], and promote M2 microglia response and functional recovery after ICH. In addition to IL-4 and IL-10, low-density lipoprotein receptor-related protein-1 (LRP1) also enhanced the M2 polarization of microglia after acute brain injury and improved neurological injury [47]. In addition, concomitant administration of cyclic adenosine monophosphate (cAMP) and IL-4 resulted in upregulation of Arg-1 expression, a phenotypic marker of M2 in mice with TBI, and also reduced reactive oxygen species (ROS) production in mice [48]. In collagenase and blood-induced ICH models, M2-type marker levels change differently. For example, most M2 markers such as IL-1 receptor antagonist [49], IL-10 [50], Arg-1, YM1, and CD206 mRNA [46] showed elevated expression levels within one day after collagenase-induced ICH, whereas TGF-β1, IL-4 mRNA, and protein expression levels showed a significant increase until three days after ICH [21,49]. In the blood-induced ICH model, IL-13 expression levels were upregulated only in the first three days and TGF-β1 in the first ten days, while IL-10 was unchanged in the first two weeks, and in addition, IL-4 expression was not detected in the perihematomal brain tissue [26].

Peroxisome proliferator-activated receptor-γ (PPARγ) belongs to the nuclear receptor superfamily and plays an important role in antioxidant and anti-inflammatory responses [51]. The binding of PPARγ to DNA in nuclear extracts was inhibited one hour after blood-induced ICH in rats. At the same time, PPAR-γ agonists were found to reduce the activation of NF-κB [52] and decrease the levels of M1 microglia pro-inflammatory factors, such as iNOS, TNF, and IL-1 [53], in addition to promoting microglia in vitro by inducing the expression of CD36 protein phagocytosis of erythrocytes in vitro, which contributes to hematoma clearance [54]. Studies have shown that statins can reduce neuroinflammation by activating PPAR while interfering with NF-κB activity and downregulating the expression of pro-inflammatory cytokines such as IL-6 and IL-23, in addition to promoting the secretion of IL-4 and the polarization of the M2 phenotype [55].

Serine/threonine protein kinase (mTOR) signaling regulates immune responses in many neurological diseases, including traumatic brain injury and Alzheimer’s [56]. Inhibition of mTOR signaling reduces deleterious microglial activity and promotes anti-inflammatory M2 polarization [57]. ICH leads to dysregulation of mTOR activation, while rapamycin and AZD8055, mTOR inhibitors, improve early brain injury by inhibiting IL-1β and TNF production [58]. In addition, in collagenase-induced ICH rats, phosphorylation of mTOR was significantly increased within 30 min, while rapamycin treatment resulted in a dose-dependent increase in the expression levels of the M2 microglia markers IL-10 and TGF-β and the ratio of IL-10 to IFNγ [59]. In addition, rapamycin intervention has been shown to reduce the expression levels of TNF, IL-1β, and IL-6 [60], and this evidence suggests a link between mTOR inhibition and M2 microglia (Figure 1), (Table 1).

Activation of high-mobility histone 1 (HMG1) and Toll-like receptor (TLR)2 or TLR4 promotes M1-like responses in microglia via the NF-κB pathway. Binding of IFNγ to the receptor promotes microglia polarization to the M1 state. STAT6 accumulates in response to IL-4 and is responsible for transcription of M2-associated genes. IL-13 and IL-4 upregulate the expression of M2 microglia markers, including Arg-1 and YM1, in microglia via the JAK-STAT6 pathway. IL-10 inhibits the production of anti-inflammatory factors. Sphingosine-1-phosphate (S1P) receptor signalling contributes to the downregulation of proinflammatory cytokines and enhances M2-like responses after ICH; MYD88, myeloid differentiation primary response protein MYD88; TRIF, TIR domain-containing adaptor molecule 1; PPARγ, Peroxisome proliferator-activated receptor-γ.

## 3. Therapeutic Targets for Microglia Polarization

When developing targeted therapies to alter microglia polarization, it is important to consider the range of targets available, such as enzymes, cell surface markers, transcription factors, and signaling proteins. Because these affect the inflammatory signaling cascade induced in M1 or M2 microglia, strategies to mitigate inflammatory damage due to brain hemorrhage are very important. Over the years, many bioactive drugs derived from different natural resources have been identified as effective therapeutic approaches. In addition, previous studies have shown that upregulation of M2 microglia expression by natural products or compounds effectively reduces neuroinflammatory responses. Therefore, developing compounds that modulate M1/M2 polarization has been considered a beneficial therapeutic strategy for neurological diseases.

### 3.1. Enzymes as Targets

#### 3.1.1. AMP-Activated Protein Kinase

AMP-activated protein kinase (AMPK) is a metabolism-sensitive serine/threonine protein kinase involved in transitioning from a pro-inflammatory M1 phenotype to an M2 phenotype. It plays a central role in regulating the pathogenesis of neuroinflammation and central nervous system diseases [69]. Recent studies have shown that activation of AMPK significantly promotes macrophage polarization toward the M2 phenotype to suppress the inflammatory response [70]. PPARγ regulates multiple pathways involving inflammation, carbohydrate, and lipid metabolism. In addition, PPARγ activation inhibits inflammatory responses and plays an important role in neuroprotection. The angiotensin II receptor blocker (ARB) telmisartan has potent neuroprotective effects in neurodegenerative diseases, promoting M2 polarization and decreasing M1 polarization in endotoxin-stimulated BV2 and primary microglia, the effects of which are partially dependent on PPARγ activation; in addition, AMPK inhibitors or AMPK knockdown attenuate the promoting effect of telmisartan on M2 polarization. It was shown that telmisartan enhanced brain AMPK activation and M2 gene expression in a mouse model of lipopolysaccharide-induced neuroinflammation. Although different functions activate AMPK and PPARγ pathways, they are both involved in lipopolysaccharide-induced M2 polarization in microglia. Both reduce the expression of inflammatory genes and protect energy metabolism [71]. Quercetin is a natural polyphenol with several biological properties, including antioxidant and anti-inflammatory effects [72], and also exerts neuroprotective effects in neurodegenerative diseases [73]. Quercetin not only decreased the expression of M1 markers such as IL-6, TNF-α, and IL-1β in microglia but also decreased M1 polarization-related chemokines. In addition, quercetin increases the levels of M2 marker IL-10 and endogenous antioxidant heme oxygenase (H2O)-1 through the activation of AMPK and protein kinase B(AKT) signaling pathways. It is expected to be a potential drug for treating inflammatory diseases of the central nervous system [74].

#### 3.1.2. Matrix Metalloproteinase (MMP) 3/9

Matrix metalloproteinase (MMP) 3/9 may disrupt the blood–brain barrier and play an important role in ICH. *Sinomenine* is an active alkaloid extracted from the plant Sinomenine acutum, which produces an anti-inflammatory response and modulates the immune system [75]. Sinomenine inhibits microglia infiltration and activation in vivo and in vitro and mediates apoptosis of hippocampal neurons by inhibiting the caspase-3 activity of microglia. In addition, Sinomenine attenuated MMP-3/9 expression, brain water content, and neurological damage in ICH, suggesting that Sinomenine is an immunomodulator of microglia polarization, inhibits microglia M1 polarization, and promotes M2 polarization, contributing to inflammation regulation and cerebral protection, providing new clues for the potential treatment of ICH [76,77].

#### 3.1.3. Janus-Activated Kinase

A Janus-activated kinase (JAK)-STAT pathway regulates cell proliferation, immunity, apoptosis, and inflammation [78]. Erythropoietin (EPO) is a pleiotropic cytokine reported to prevent neuronal apoptosis in many cerebrovascular diseases [79,80]. Janus family tyrosine-protein kinase JAK2 plays a central role in erythropoietin receptor (EPOR) downstream signaling. At the same time, phosphorylation of STAT3 is located downstream of p-JAK2. EPO amplifies the oxyhemoglobin (OxyHb)-induced increase in p-JAK2 and p-STAT3, while the P JAK2 inhibitor AZD1480 blocked the EPO-stimulated-induced upregulation of p-STAT3. It was shown that EPO decreased the gene expression of pro-inflammatory cytokine genes (tumor necrosis factor-α and IL-1β) and increased the expression of anti-inflammatory cytokine genes (IL-4 and IL-10) in vivo and in vitro. However, EPOR knockdown and AZD1480 reversed the EPO-mediated changes in pro-inflammatory cytokine gene and anti-inflammatory cytokine gene expression after OxyHb stimulation, i.e., EPO may have promoted microglia M2 polarization through activation of the JAK2/STAT3 pathway [81,82].

### 3.2. Targeting Protein Receptors

#### 3.2.1. Tropomyosin-Related Kinase

Tyrosine kinase receptors of the Trk family (TrkA, TrkB, TrkC) play a key role in the recovery after brain injury [83]. The TrkB/brain-derived growth factor (BDNF) pathway is widely recognized as a key pathway for repair after brain injury [84]. Minocycline exerts neuroprotective effects by promoting the secretion of neurotrophic factors from M2 microglia through the TrkB/BNDF pathway [85]. *Minocycline* is a semi-synthetic second-generation tetracycline derivative that readily penetrates the blood–brain barrier and is widely used to reduce bacterial load and inflammation [86]. It was shown that minocycline decreased the proliferation of CD68+ cells and increased the number of arginase 1 + CD16+ cells induced by ICH, suggesting that minocycline promotes the conversion of M1 microglia to M2 type after ICH in rats. In addition, minocycline significantly increased the expression of M2 microglia-derived BDNF around neuronal cells [85].

#### 3.2.2. Peroxisome Proliferator-Activated Receptor-γ

Peroxisome proliferator-activated receptor-γ (PPARγ) is a member of the nuclear hormone receptor family and is thought to be involved in regulating a variety of metabolic, endocrine, and cardiovascular diseases [87]. Rosiglitazone, a PPARγ agonist, significantly improves the pathological changes in the brain’s white matter after stroke while protecting the white and gray matter of the brain and promoting long-term functional recovery after stroke. In vitro and in vivo studies have shown that rosiglitazone exerts protective effects on neurons, reduces oxidative stress, and attenuates excitotoxicity by promoting endogenous oligodendrocyte differentiation and microglia differentiation toward the M2 phenotype (including Arg1, IL-10) [88,89]. 10-O-(N, N dimethylaminoethyl)-ginkgolide B methanesulfonate (XQ-1H) is a new derivative of ginkgolide B with a strong platelet-activating factor antagonistic effect [90]. It was shown that co-expression of CD16 and Iba1 was significantly reduced, and co-expression of CD206 and Iba1 was increased in response to XQ-1H, indicating that XQ-1H affects microglia polarization by activating the PPARγ signaling pathway to promote the anti-inflammatory phenotype (CD16) and inhibit the pro-inflammatory phenotype (CD206) [91].The herbal chemical constituent of forsythia, forsythoside, has various biological functions, such as improving islet resistance [92], regulating apoptosis and oxidative stress [93], and antiviral and anti-inflammatory activities of severe acute respiratory syndrome coronavirus 2 (SARS-CoV-2) and human coronavirus 229E (HCoV-229E) have antiviral and anti-inflammatory activities [94]. It was shown that coniferin could have anti-inflammatory effects by promoting microglia M2 polarization through the PPARγ/NF-κB pathway. It could also reduce microglia-induced blood–brain barrier damage after brain injury, improving its post-ischemic tissue damage [47].

#### 3.2.3. Toll-Like Receptors

Toll-like receptor 4 (TLR4) is a key regulator of microglia activation and their polarization after brain injury [95]. Inhibition of TLR4 signaling attenuates neurological deficits by modulating the M1/M2 phenotypic shift in microglia in a mouse model of traumatic brain injury [96]. Caryophyllene (β-caryophyllene, BCP) is a natural bicyclic sesquiterpene with a variety of biological and pharmacological effects such as analgesic [97,98], anti-inflammatory [99,100], antioxidant [101], and prevention of apoptosis [102]. BCP is a small lipolytic molecule that can cross the blood–brain barrier [103]. It was found that, in wild-type mice and TLR4 knockout mice, BCP inhibited the polarization of microglia toward the M1 phenotype and promoted polarization toward the M2 phenotype. Furthermore, in vitro, BCP mediated the activation and polarization of primary microglia induced by the combination of lipopolysaccharide and interferon-γ. This effect of BCP was accompanied by downregulation of TLR4 and CD16/32 and upregulation of CD206 and was enhanced by blocking TLR4 activity. These findings suggest that TLR4 is an important target of BCP and that its protective effects are exerted, at least in part, through TLR4-mediated microglia activation and promotion of microglia polarization toward a beneficial M2 phenotype [104]. Pinocembrin (5,7-dihydroxyflavanone) is a natural product extracted from propolis with neuroprotective [105,106], anti-inflammatory, and oxidative stress-reducing effects [106]. Pinocembrin reduces the number of classically activated M1-like microglia and decreases the activation of M1-associated cytokines and markers (IL-1β, IL-6, TNF-α, and iNOS), NF-κB, and the expression of TLR4 and its downstream target proteins TIR domain-containing adaptor molecule 1(TRIF) and MyD88. Moreover, inhibition of the TLR4 signaling pathway and reduction of M1-like microglia polarization may be the main mechanism of pinocembrin protection against hemorrhagic brain injury. These findings suggest that the pinocembrin may potentially treat brain hemorrhage and other acute brain injuries [25].

### 3.3. Targeting Transcription Factors

#### 3.3.1. PPARγ Coactivator-1α

PPARγ coactivator-1α (PGC-1α) is a transcriptional co-activator of nuclear receptors that plays a key integrative role in the transcriptional regulation of cellular energy metabolism, oxidative stress defense, mitochondrial function, and biogenesis [107,108]. A recent study developed that PGC-1α inhibits endotoxin-induced M1 activation by suppressing NF-κB activity and promotes microglia polarization to the M2 phenotype by activating the STAT6 and STAT3 pathways [109]. Resveratrol is a natural polyphenol with pharmacological effects such as anti-inflammatory, anti-apoptotic, and antioxidant effects, in addition to penetrating the blood–brain barrier, inhibiting the activation of glial cells, and reducing the production of pro-inflammatory factors [110]. Studies have shown that resveratrol activates silent information regulator-1 (SIRT1), upregulates PGC-1α expression, attenuates inflammatory damage, and promotes microglia differentiation to the M2 phenotype [109].

#### 3.3.2. Nuclear Factor-κB

Nuclear factor-κB (NF-κB) is a conventional transcription factor activated by lipopolysaccharide and regulates the expression of most M1 marker genes (genes for pro-inflammatory cytokines). Anisol (p-methoxybenzyl alcohol, PMBA) is isolated from the natural medicine Gastrodia [111], PMBA significantly reduced lipopolysaccharide-induced production of tumor necrosis factor-alpha, prostaglandin E2 (PEG-2), and nitric oxide without cytotoxicity. In addition, it increased the levels of anti-inflammatory factor IL-10 and TGF-β. Phenotypic analysis after lipopolysaccharide stimulation of microglia showed that PMBA significantly downregulated the expression of the M1 microglia marker CD16/32 and upregulated the expression of the M2 microglia marker CD206, with the possible mechanism of reducing the production of inflammatory mediators and cytokines by inhibiting the activation of NF-κB and mitogen-activated protein kinase (MAPK) [112]. Fingolimod (FTY720) is a sphingosine-1-phosphate (S1P) receptor one antagonist, and studies have shown that S1P1 significantly reduces the inhibitory effect of FTY720 on the production of tumor necrosis factor-α by activated microglia, suggesting that FTY720 inhibits the production of pro-inflammatory cytokines by microglia via SIP1 [113]. In contrast, the signaling pathway downstream of SIP1 inhibits NF-κB activation by downregulating histone deacetylase (HDAC), leading to the downregulation of pro-inflammatory cytokines and promoting the expression of neurotrophic factors with neuroprotective effects. In addition, FTY720 can control important inflammatory gene targets by regulating STAT1 levels at promoter sites, thereby inhibiting STAT1 autophagy and converting pro-inflammatory microglia into anti-inflammatory microglia [114]. Tanshinone IIA is a lipid-soluble diterpenoid isolated from Salvia miltiorrhiza, which has been shown in several studies to have neuroprotective effects against cerebral ischemic injury through inhibition of inflammation and autophagy [115]. Studies have shown that tanshinone IIA inhibits M1 microglia polarization and promotes M2 cell polarization through the NF-κB pathway, thereby suppressing the inflammatory response [116]. Baicalein (5,6,7-trihydroxyflavonoids) is the main bioactive component extracted from the roots of Scutellaria baicalensis, a commonly used herbal medicine. Baicalein significantly decreased the expression of M1 markers CD16 and CD86 and increased the expression of M2 markers CD163 and CD206, suggesting that baicalein inhibits M1 polarization and promotes microglia/macrophage M2 polarization, thereby suppressing neuroinflammation. In addition, baicalein inhibits NF-κB signaling by reducing the phosphorylation and nuclear translocation of IκBα, thereby reducing the release of pro-inflammatory factors IL-6, IL-18, and TNF-α [117].

### 3.4. Targeting Inflammatory Vesicles

#### NLRP3 Infammasomes

NLRP3 infammasomes is a multi-protein complex that regulates the maturation and secretion of pro-inflammatory cytokines, including IL-1β and IL-18. Paeonol (2’-hydroxy-4’-methoxyacetophenone), the main active ingredient in peony root extract, was found to significantly inhibit NLRP3 inflammatory vesicle-associated protein levels in vivo and in vitro, reverse the effects of LPS/ATP on TLR4, MYD88, and p-p65/p65 protein levels and promote M2 polarization and inhibit M1 polarization in BV-2 cells, thereby suppressing the release of inflammatory cytokines [118]. Edaravone (EDA) shifts the M1 pro-inflammatory phenotype of microglia to an M2 anti-inflammatory state by decreasing the expression of M1 markers (TNFα and IL-1β) and promoting the expression of M2 markers (Arg-1 and IL-10). Furthermore, EDA suppressed the inflammatory response by inhibiting the expression of pro-inflammatory factors IL-1β, IL-18, and NO, but the neuroprotective effect of EDA was ineffective in the presence of siRNANLRP3, suggesting that EDA may exert anti-inflammatory effects by inhibiting NLPR3 inflammatory tissue activation and regulating microglia M1/M2 polarization [119] (Table 2).

## 4. Conclusions

Appropriate anti-brain hemorrhage therapy should promote the right microglia phenotype at the right time, thus maximizing the natural process of hematoma clearance and brain repair. Some drugs have now been validated in basic studies to have a reparative effect on secondary damage after ICH, where inhibition of M1-like microglia activation and enhancement of M2-like microglia anti-inflammatory response are the main mechanisms of action of these compounds. The next step in the search for therapies that selectively promote anti-inflammatory microglia phenotypic polarization is clinically important to improve the prognosis of ICH. In addition, existing studies suggest that interactions between microglia and astrocytes and oligodendrocytes may be beneficial or detrimental to neurons, but the crosstalk between the three has not been fully investigated. Further studies of astrocytes and oligodendrocyte mediators that regulate microglia polarization and phagocytosis will improve our understanding of the pathogenesis of ICH and lead to more effective therapeutic options for patients with ICH.

## Figures and Tables

**Figure 1 molecules-27-07080-f001:**
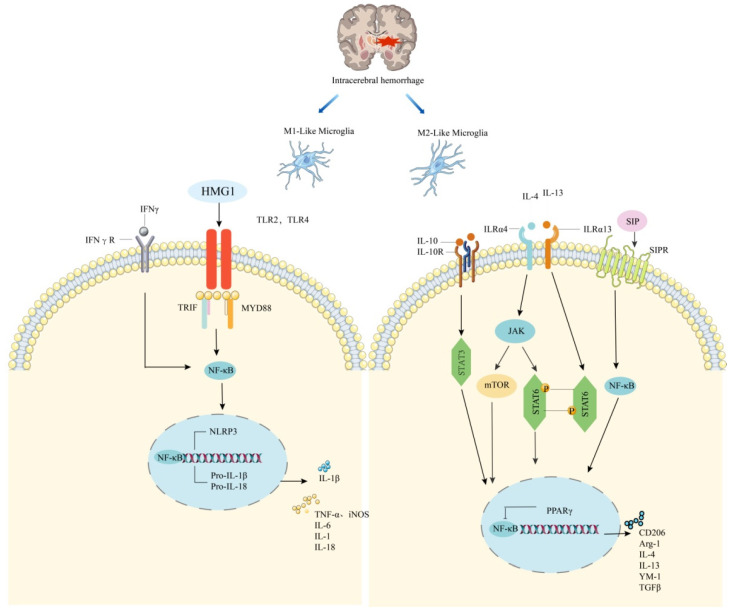
Mechanism of microglia polarization after cerebral hemorrhage.

**Table 1 molecules-27-07080-t001:** Markers and function of M1 and M2 microglia.

Phenotype	Marker	Type	Function	References
M1	IL-1β	Cytokine	Proinflammatory	[18,21,30]
	IL-6	Cytokine	Proinflammatory	[21]
	TNF	Cytokine	Proinflammatory	[21]
	IFN-γ	Cytokine	Proinflammatory, M1 microglia, and macrophage inducer	[23,24]
	iNOS	Metabolic enzyme	Mediates nitric oxide synthesis	[23,24]
	CD16	Immunoglobulin Fc receptor	Induces proinflammatory signalling	[25]
	CD32	Immunoglobulin Fc receptor	Induces proinflammatory signalling	[25]
	CD86	Surface receptor	Classic M1 microglia and macrophage marker	[61]
	CCL5	Chemokine	Recruits immune cells	[62]
	CCL20	Chemokine	Recruits immune cells	[63]
	CXCL1	Chemokine	Recruits immune cells	[64]
	CXCL10	Chemokine	Recruits immune cells	[65]
	MHC-Ⅱ	Surface receptor	Mediates T cell differentiation to Th1	[66]
M2	IL-4	Cytokine	Anti-inflammatory, increases microglia and macrophage phagocytosis	[42]
	IL-10	Cytokine	Anti-inflammatory, mediates microglia and macrophage phagocytosis	[50]
	IL-13	Cytokine	Anti-inflammatory	[42]
	TGF-β	Cytokine	Anti-inflammatory	[26,46]
	Arg-1	Cytosolic enzyme	Suppresses inflammation; upregulated by IL-4 and IL-13	[48]
	YM1	Secretory protein	Anti-inflammatory; induction depends on IL-4 and IL-13	[46]
	CCL22	Chemokine	Recruits dendritic cells, Th2 cells and regulatory T cells	[67]
	CD163	Scavenger receptor	Haemoglobin clearance	[68]
	CD206	Mannose receptor	Mediates endocytosis and phagocytosis in response to microglia and macrophage activation	[46]

Arg-1, arginase 1; CCL, chemokine (CC motif) ligand; CD206, macrophage mannose receptor 1; CXCL, chemokine (CXC motif) ligand; IFN-γ, interferon gamma; iNOS, inducible nitric oxide synthase; MHC-Ⅱ, major histocompatibility complex-I; TGF-β, transforming growth factor-β; TNF, tumour necrosis factor; YM1, chitinase like protein 3.

**Table 2 molecules-27-07080-t002:** Drugs that promote microglia polarization.

Type	Target	Drug	Effects on Microglia	References
Enzymes	AMPK	Telmisartan	Decreases M1-like microglial responses Enhances M2-like microglial responses	[71]
		Quercetin	Decreases M1-like microglial responses Enhances M2-like microglial responses	[74]
	MMP3/9	Sinomenine	Decreases M1-like microglial responses Enhances M2-like microglial responses	[76,77]
	JAK	Erythropoietin	Decreases M1-like microglial responses Enhances M2-like microglial responses	[81,82]
Protein receptors	Trk	Minocycline	Promotes M1-to-M2 phenotype shift	[85]
	PPAR-γ	Rosiglitazone	Enhances M2-like microglial responses	[88,89]
		HQ-1H	Decreases M1-like microglial responses Enhances M2-like microglial responses	[91]
		Forsythoside	Enhances M2-like microglial responses	[47]
	TLR4	Caryophyllene	Decreases M1-like microglial responses Enhances M2-like microglial responses	[104]
		Pinocembrin	Decreases M1-like microglial responses	[25]
Transcription factors	PGC-1α	Resveratrol	Enhances M2-like microglial responses	[109]
	NF-κB	Anisol	Decreases M1-like microglial responses Enhances M2-like microglial responses	[112]
		Fingolimod	Decreases M1-like microglial responses Promotes M1-to-M2 phenotype shift	[114]
		Tanshinone IIA	Decreases M1-like microglial responses Enhances M2-like microglial responses	[116]
		Baicalein	Decreases M1-like microglial responses Enhances M2-like microglial responses	[117]
Inflammatory vesicles	NLRP3	Paeonol	Decreases M1-like microglial responses Enhances M2-like microglial responses	[118]
		Edaravone	Promotes M1-to-M2 phenotype shift	[119]

AMPK, AMP-activated protein kinase; JAK, Janus-activated kinase; MMP3/9, Matrix metalloproteinase (MMP) 3/9; NF-κB, Nuclear factor-κB; NLRP3, NOD-like receptor thermal protein domain associated protein 3; PGC-1α, PPARγ coactivator-1α; PPAR-γ, Peroxisome proliferator-activated receptor-γ; TLR4, Toll-like receptor 4; Trk, Tropomyosin-related kinase.

## Data Availability

Not applicable.

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
