# Peer review of "Mechanism and Regulation of Microglia Polarization in Intracerebral Hemorrhage"

_molecules, 2022, doi:10.3390/molecules27207080_

Round 1
Reviewer 1 Report
This is another review article about the role of microglia in intracerebral hemorrhage, with special attention being given to microglia polarization. While I am not a priori against the concept of polarization, I would rather subscribe to a moderate view of seeing various microglia phenotypes rather as a spectrum then the existance of 2 separate "poles". Main problem is discerning microglia, as resident brain cells from macrophages, which upon disruption of blood-brain barrier invade brain tissue. While polarization of macrophages to M0,1,2 phenotypes is universaly accepted, such division among microglia is dubious, which authors acknowledge in the text. I still think it should be emphasized earlier, perhaps already in the abstract.
Several minor problems:
Abstract - microglia a, as glial cells are not "non-neural", they belong together with neuronal cells and other glial cells to "neural cells". I suggest to erase the "non-neural" qualification. Further, it should be eplhasized, perhaps already in the abstract that much of the research done and presented here would not discern resident microglia cells from the macrophages invading brain tissue.
I do not understand the sentence "The medium affects the brain's repair process.", I suggest to delete it, or explain better.
Otherwise, it is an interesting review worth publishing.
Author Response
Response to Reviewer 1 Comments
Dear reviewer:
On behalf of my co-authors, we thank you for giving us a chance to revise and improve the quality of our article. We have read your comments carefully and have tried our best to revise our manuscript according to the comments. Attached the revised version, which we would like to submit for your kind consideration. Here, we would like to show the changes briefly as follows:
Point 1: Abstract - microglia a, as glial cells are not "non-neural", they belong together with neuronal cells and other glial cells to "neural cells". I suggest to erase the "non-neural" qualification.
Response 1: Thank you for your patient review and comments. In lines 21-23 of the abstract, the explanation that microglia are "non-neural cells" has been removed. The original " In the innate immune response to brain hemorrhage, microglia are the first non-neural cells to appear around the injured tissue and are involved in the inflammatory cascade response." was changed to " In the innate immune response to cerebral hemorrhage, microglia first appear around the injured tissue and are involved in the inflammatory cascade response." Thank you for your patient review.
Point 2: It should be eplhasized, perhaps already in the abstract that much of the research done and presented here would not discern resident microglia cells from the macrophages invading brain tissue.
Response 2: Thank you for your patient review and comments. Activated microglia and recruited macrophages after intracerebral hemorrhage show some common characteristics and cannot be distinguished. The type of polarization of microglia is similar to that of macrophages, which is explained in the revised article (The contents of 60-73 lines of yellow marks). Thank you for your patient review.
Point 3: I do not understand the sentence "The medium affects the brain's repair process.", I suggest to delete it, or explain better.
Response 3: Thank you for your patient review and comments. This sentence has been deleted from the article. Thank you for your patient review.
Reviewer 2 Report
In the current work, the authors aimed at discussing the mechanisms and regulation of microglia polarization in intracerebral hemorrhage. They also stated that they aimed to discuss the compounds and natural pharmaceutical ingredients that can polarize the M1 to the M2 phenotype. However, the manuscript is very poorly written. It is mainly composed of extremely long unorganized paragraphs in which sentences are loosely connected. Additionally, the second aim (i.e. to discuss the compounds and natural pharmaceutical ingredients that can polarize the M1 to the M2 phenotype) was almost forgotten throughout the manuscript. Only table 2 listed some drugs that promote microglia polarization (without even referring to the reference of each). Besides, the language, format, and abbreviations usage are all inappropriate and require extensive revision.
Author Response
Dear reviewer:
On behalf of my co-authors, we thank you for giving us a chance to revise and improve the quality of our article. We have read your comments carefully and have tried our best to revise our manuscript according to the comments. Attached the revised version, which we would like to submit for your kind consideration. Here, we would like to show the changes briefly as follows:
Point 1: In the current work, the authors aimed at discussing the mechanisms and regulation of microglia polarization in intracerebral hemorrhage. They also stated that they aimed to discuss the compounds and natural pharmaceutical ingredients that can polarize the M1 to the M2 phenotype. However, the manuscript is very poorly written. It is mainly composed of extremely long unorganized paragraphs in which sentences are loosely connected.
Response 1: Thank you for your patient review and comments. The content related to the polarization of microglia in the second paragraph of our text has been modified and integrated into the first paragraph to make the article more concise. Thank you for your patient review.
Point 2: Additionally, the second aim (i.e.to discuss the compounds and natural pharmaceutical ingredients that can polarize the M1 to the M2 phenotype) was almost forgotten throughout the manuscript. Only table 2 listed some drugs that promote microglia polarization (without even referring to the reference of each).
Response 2: Thank you for your patient review and comments. We supplement the references on the two phenotypic markers of microglia in the first table and the references in the second table on compounds and natural medicines that promote M1 to m2 polarization. Thank you for your patient review.
Point 3: Besides, the language, format, and abbreviations usage are all inappropriate and require extensive revision.
Response 3: Thank you for your patient review and comments. We modified the subheadings of lines 91, 211, 223, 249, 261, 277, 289, 313, 340, 353 and 388 to make the article logical, and we modified inappropriate acronyms, language and format to make the article more fluent. Thank you for your patient review.
Round 2
Reviewer 2 Report
The manuscript has been improved, however, I have some minor comments:
1. I don't think that a discussion is necessary in this review. The first paragraph of the discussion can be removed to the introduction (references of the data in this paragraph are required). The second paragraph can stand as a conclusion.
2. Table 1 and table 2 are so confusing. Borders should be added to separate rows of M1 and M2 in table 1, and to separate the drugs acting on each target and the targets classified under each each type in table 2.
3. Footnotes that define the abbreviations in the tables are needed.
4. Further revision of the language, punctuation, use of abbreviations and format is necessary.
Author Response
Dear reviewer:
On behalf of my co-authors, we thank you for offering us an opportunity to improve the quality of our submitted manuscript. We have read your comments carefully and have tried our best to revise our manuscript according to the comments. Attached the revised version, which we would like to submit for your kind consideration. We highlighted the revisions in red colour. Here, we would like to show the changes briefly as follows:
Point 1: I don't think that a discussion is necessary in this review. The first paragraph of the discussion can be removed to the introduction (references of the data in this paragraph are required). The second paragraph can stand as a conclusion.
Response 1: Thank you for your patient review and comments. We moved the first paragraph of the discussion to the introduction and put the second paragraph in the last paragraph as a conclusion. Thank you for your patient review.
Point 2: Table 1 and table 2 are so confusing. Borders should be added to separate rows of M1 and M2 in table 1, and to separate the drugs acting on each target and the targets classified under each each type in table 2.
Response 2: Thank you for your patient review and comments. We added a border in Table 1 to separate M1 and M2. And borders have been added in Table 2 to separate the drugs acting on each target and the targets classified under each each type. Thank you for your patient review.
Point 3: Footnotes that define the abbreviations in the tables are needed.
Response 3: Thank you for your patient review and comments. Footnotes have been added at the bottom of tables 1 and 2, respectively. Thank you for your patient review.
Point 4: Further revision of the language, punctuation, use of abbreviations and format is necessary.
Response 4: Thank you for your patient review and comments.Footnotes. We modified inappropriate acronyms, language and format to make the article more fluent. Thank you for your patient review.
